# Spatial Modelling to Inform Public Health Based on Health Surveys: Impact of Unsampled Areas at Lower Geographical Scale

**DOI:** 10.3390/ijerph17030786

**Published:** 2020-01-28

**Authors:** Kevin Watjou, Christel Faes, Yannick Vandendijck

**Affiliations:** Interuniversity Institute for Biostatistics and Statistical Bioinformatics, Hasselt University, 3500 Hasselt, Belgium; christel.faes@uhasselt.be (C.F.); yannick.vandendijck@uhasselt.be (Y.V.)

**Keywords:** model-based inference, small area estimation, spatial smoothing, survey weighting, missing areas

## Abstract

Small area estimation is an important tool to provide area-specific estimates of population characteristics for governmental organizations in the context of education, public health and care. However, many demographic and health surveys are unrepresentative at a small geographical level, as often areas at a lower level are not included in the sample due to financial or logistical reasons. In this paper, we investigated (1) the effect of these unsampled areas on a variety of design-based and hierarchical model-based estimates and (2) the benefits of using auxiliary information in the estimation process by means of an extensive simulation study. The results showed the benefits of hierarchical spatial smoothing models towards obtaining more reliable estimates for areas at the lowest geographical level in case a spatial trend is present in the data. Furthermore, the importance of auxiliary information was highlighted, especially for geographical areas that were not included in the sample. Methods are illustrated on the 2008 Mozambique Poverty and Social Impact Analysis survey, with interest in the district-specific prevalence of school attendance.

## 1. Introduction

Demographic and health surveys can provide good information about the geographical distribution of indicators related to education, public health and care. Such surveys often follow a complex sampling design. Two approaches are commonly used to model such survey data: Design-based and model-based methods, where the latter can be subdivided into area-level and unit-level models. The most well-known design-based estimator is the Horvitz–Thompson estimator, a weighted estimator which provides unbiased estimates when working with complex surveys [1]. These estimators provide inferences based on the randomization distribution of the collected sample. Lately, more effort has been put into the development of model-based estimators, and more specifically hierarchical (or random effects) spatial smoothing models [2,3,4,5].

Most demographic and health surveys are designed to obtain representative samples at the national or regional level. However, at lower geographical areas, the sample is unrepresentative; as only a subset of the small areas is being sampled. This implies that direct estimates are only available in these areas where samples were taken, i.e., the in-sample areas. Estimates in areas that are not sampled, i.e., the off-sample areas, must be obtained by indirect estimation using methods borrowing information from the available in-sample areas. When the number of off-sample areas is large, obtaining reliable estimates in these areas may be problematic because information needs to be borrowed from neighboring areas and not enough in-sample neighbors are available.

In this paper, interest is in the performance of model-based small area estimation (SAE) methods in the presence of many off-sample areas. The focus is on a binary health outcome obtained from a complex survey. It is assumed that design weights are available for all sampled individuals. A comparison of several methods which were recently proposed in the SAE literature is made to estimate the prevalence of the health outcome at a small geographical scale, and the impact of missing information at this geographical scale is investigated. Furthermore, in order to obtain reliable estimates for these off-sample areas, we propose a model where the survey data is augmented with auxiliary summary data from a higher administrative level. The performances of these models are evaluated using several summary statistics, which measure the precision and accuracy of the estimates: Squared bias, variance, mean squared error (MSE), coverage probabilities for the nominal 95% confidence and credible intervals and the average length of these intervals. Note that here we only consider the problem of data that are missing by design of the survey. While non-response of participants is not addressed here, many literature can be found on this subject [4,5,6,7].

As a motivating example, we investigate the performance of the described methods using the 2008 Education PSIA (Poverty and Social Impact Analysis) survey of Mozambique. While the survey is designed to investigate estimates at the national, urban/rural, and provincial level, we consider the spatial distribution at the smaller district division of Mozambique. In this survey no less than 65 of the 125 district areas are off-sample. First of all, this survey is used to investigate the performance of the described methods through an extensive simulation study. Secondly, we estimate the prevalence of current or historical school attendance in the districts. Lastly, we introduce a method which can provide us with more reliable estimates in the off-sample areas by means of auxiliary information. For this paper, we look at the primary energy source of the households as a surrogate indicator of wealth.

The structure of the paper is as follows. In Section 2, the 2008 Mozambique PSIA Survey is introduced as the motivating example. In Section 3, the methodology is described, including the notation which was used throughout the manuscript, the design-based estimators, the model-based estimators and models incorporating auxiliary data. In Section 4, a simulation study was conducted to evaluate the different methods. Lastly, results of the analyses of school attendance, based on the 2008 Mozambique PSIA Survey are given in Section 5.

## 2. The 2008 Mozambique PSIA Survey

The 2008 Mozambique Poverty and Social Impact Analysis Survey was designed to investigate school enrollment rates in Mozambique. A stratified multi-stage sampling methodology was applied, which entailed selection of enumeration areas (EAs) stratified by province, urban, and rural region, followed by the selection of households within the EA. Mozambique is divided into ten provinces and one city (Maputo City). These are in turn subdivided into 125 districts and 858 enumeration areas (AE). The survey ensured representative samples for national, urban/rural, and regional estimates. In total, 221 enumeration areas were sampled (86 urban strata and 135 rural strata). At the next sampling stage 12 households were selected within each sample urban EA and nine households within each sample rural EA, using random systematic sampling with equal probability. The resulting PSIA sample consists of 7836 observations related to the school attendance of the children of the household. In total, 5873 (74.95%) answered “Yes” and 1963 (25.05%) answered “No” to the question of interest.

Note that the level of the enumeration areas is the spatial level at which the sampling scheme was conducted. The performed analyses, which will be discussed in the next sections, are performed at the spatial level of the districts, which are comprised of multiple enumeration areas. The geographical distribution of the district-specific sample sizes is visualized in Figure 1. Note that the small areas which are highlighted in white are the off-sample areas in the PSIA sample. The percentage of off-sample area is 52% (65/125). The area highlighted in blue in Figure 1 in the northern part of Mozambique represents part of Lake Malawi.

A more in-depth summary and further details on the entire sampling scheme for the PSIA study can be found at http://microdata.worldbank.org/index.php/catalog/999/download/20597.

### Auxiliary Data

The auxiliary data is related to the sustainable development goal 7 to ensure access to affordable, reliable, and modern energy. The data describes the primarily access to different energy sources (electricity, generator/solar panel, gas, oil/kerosene/paraffin, candle, battery, firewood, or other). We summarize the information on the proportion of households primarily using a specific type of energy source per district in Table 1. The majority of the districts primarily used firewood and oil-based fuels as household energy sources. On the other hand, electricity was more common in the more populated districts. The other energy sources appear scarcely in the population.

Population data for each district is available at the site of the National Institute of Statistics of Mozambique: (http://www.ine.gov.mz/estatisticas/estatisticas-territorias-distritais).

## 3. Methodology

Let $Y_{ik}$ be a binary health outcome for individual *i* in small area *k* ($i=1,\ldots ,N_{k}$ and k=1,…,K) with $N_{k}$ the population size in area *k*. We assume that $N_{k}$ is known for each area. A sample of size $n_{k}$ is drawn from each area *k*, where some of the $n_{k}$ could be zero. Denote the sampled values by $y_{ik}$. Let $N=\sum_{k=1}^{K}N_{k}$ and $n=\sum_{k=1}^{K}n_{k}$ represent the total population and sample size, respectively. We focused on estimating the true prevalence, $P_{k}$, in each area *k*, namely
(1)$$P_{k}=\frac{1}{N_{k}}\sum_{i=1}^{N_{k}}Y_{ik}.$$

Let $R_{ik}$ denote the binary variable indicating whether the $i^{th}$ individual in area *k* is sampled $\left(R_{ik}=1\right)$ or not $\left(R_{ik}=0\right)$. We use $s_{k}$ to indicate the set of sampled individuals in area *k* and $s_{k}^{\prime }$ for those that are not sampled.

To reflect the sampling design, weights are attached to each respondent’s outcome. These weights are denoted as $w_{ik}$ for an individual *i* in district *k*. These weights can reflect both or a combination of the complex survey design and post-stratification adjustments, and are proportional to the inverse probability of inclusion in the sample for unit *i* in area *k*. We further assume that all sampled individuals respond to the survey. We use normalized weights, which sum up to the sample size $n_{k}$ in area *k* and are denoted by ${\tilde{w}}_{ik}$:(2)$${\tilde{w}}_{ik}=n_{k}\frac{w_{ik}}{\sum_{i\in s_{k}}w_{ik}}.$$

### 3.1. Design-Based Methods

The most simple estimator for the population prevalence is the unweighted estimator (UNW),
(3)$${\hat{p}}_{k}^{UNW}=\frac{1}{n_{k}}\sum_{i\in s_{k}}y_{ik},$$
which corresponds to the (unweighted) sample prevalence. While this estimator is unbiased in the context of simple random sampling, it will produce biased results when the sampling scheme has a complex nature. For more complex sampling schemes, the design weights have to be taken into account in the estimation process. A common design-based estimator which takes these weights into account is the Horvitz–Thompson (HT) estimator [1]:(4)$${\hat{p}}_{k}^{HT}=\frac{\sum_{i=1}^{N_{k}}R_{ik}{\tilde{w}}_{ik}y_{ik}}{\sum_{i=1}^{N_{k}}R_{ik}{\tilde{w}}_{ik}}=\frac{1}{n_{k}}\sum_{i\in s_{k}}{\tilde{w}}_{ik}y_{ik},$$
which has variance
(5)$$\hat{\mathrm{var}}\left({\hat{p}}_{k}^{HT}\right)=\frac{1}{n_{k}}\left(1-\frac{n_{k}}{N_{k}}\right)\frac{1}{n_{k}-1}\sum_{i\in s_{k}}{\tilde{w}}_{ik}^{2}{\left(y_{ik}-{\hat{p}}_{k}^{HT}\right)}^{2}.$$

The HT estimator is a design-unbiased estimator of $P_{k}$. It is a so-called direct estimator because it uses only the responses from the area of interest [8]. The estimator has very good properties if interest is in estimation of the prevalence at a higher geographical scale, however most surveys are not designed to yield appropriate direct estimates at smaller geographical scale. Indeed, sample size in smaller areas can be too small to produce reliable or stable direct estimates. In addition, no estimates can be obtained for the non-sampled areas.

### 3.2. Model-Based Methods

As an alternative to the design-based estimator, hierarchical models are an important consideration in small area estimation [9,10,11]. In this paper, we focus on some recently investigated hierarchical spatial smoothing methods described in Mercer et al. [12]. These methods were further investigated in the context of survey data by Vandendijck et al. [3].

In general, in the context of disease mapping, the design of the survey is not taken into account. We do refer to this model as the naive binomial model (NB):(6)$$\begin{array}{ccc} y_{k}|{\tilde{p}}_{k} & ∼ & \mathrm{Binomial}(n_{k},{\tilde{p}}_{k})\\ \mathrm{logit}({\tilde{p}}_{k}) & = & \beta_{0}+u_{k}+v_{k}, \end{array}$$
where $y_{k}=\sum_{i\in s_{k}}y_{ik}$ is the total number of cases in the sample in area *k*. This is a standard model in disease mapping, in which $\beta_{0}$ is an overall effect, $u_{k}∼N\left(0,\sigma_{u}^{2}\right)$ are independent random effects taking into account extra heterogeneity amongst areas *k*, and $v_{k}$ are spatially dependent random effects. It is assumed that $v_{k}$ follows the commonly used intrinsic conditional auto-regressive (ICAR) model [13]
(7)$$v_{k}|v_{k^{\prime }}∼N\left(\frac{1}{m_{k}}\sum_{k^{\prime }\;\in \;\mathrm{ne}\left(k\right)}v_{k^{\prime }},\frac{\sigma_{v}^{2}}{m_{k}}\right),$$
where ne(k) denotes the set of neighbors of area *k* and $m_{k}$ is the number of neighbors. For identifiability reasons of the overall intercept $\beta_{0}$, the sum of the random effects $v_{k}$ is constrained to be zero [14]. In this article, we take the common approach to consider two areas as neighbors if they share a common boundary.

To account for the survey design, Mercer et al. [12] proposed to first calculate the HT estimator in the areas, and then to smooth them using a spatial smoothing model. As the HT estimators are restricted to the [0,1] interval, a transformation is applied to map them on the (−∞,+∞) interval. A first possible transformation is the empirical logistic transform. In this case, the response $y_{k}^{L}=\log\left[{\hat{p}}_{k}^{HT}/\left(1-{\hat{p}}_{k}^{HT}\right)\right]$ is modeled as
(8)$$\begin{array}{ccc} y_{k}^{L}|{\tilde{p}}_{k} & ∼ & N\left(\mathrm{logit}\left({\tilde{p}}_{k}\right),\sigma_{k}^{2}\right)\\ \mathrm{logit}({\tilde{p}}_{k}) & = & \beta_{0}+u_{k}+v_{k}, \end{array}$$
where the variance $\sigma_{k}^{2}$ is set equal to $\hat{\mathrm{var}}\left({\hat{p}}_{k}^{HT}\right)/{\left[{\hat{p}}_{k}^{HT}\left(1-{\hat{p}}_{k}^{HT}\right)\right]}^{2}$. Model (8) is further referred to as the logit-normal (LN) model. Note that the same spatial smoothing model is assumed as in (7), with both an unstructured and structured random effect. As an alternative, the arcsine square-root transformation can be used [12,15]. In this case, the response is $y_{k}^{A}=\sin^{-1}\left(\sqrt{{\hat{p}}_{k}^{HT}}\right)$, which is modeled as
(9)$$\begin{array}{r} y_{k}^{A}|{\tilde{p}}_{k}∼N\left(\sin^{-1}\left(\sqrt{{\tilde{p}}_{k}}\right),\sigma_{k}^{2}\right)\\ \sin^{-1}\left(\sqrt{{\tilde{p}}_{k}}\right)=\beta_{0}+u_{k}+v_{k}. \end{array}$$

This transformation has the advantage of stabilizing the variance, with variance $\sigma_{k}^{2}$ equal to ${\left(4n_{k}^{*}\right)}^{-1}$. In this formula, $n_{k}^{*}$ is the effective sample size $n_{k}^{*}={\hat{p}}_{k}^{HT}\left(1-{\hat{p}}_{k}^{HT}\right)/\hat{\mathrm{var}}\left({\hat{p}}_{k}^{HT}\right)$. This model is called the arcsine (AS) model.

Chen et al. [2] proposed an alternative method to take into account the sampling design. Instead of modeling the HT estimator directly, they proposed to model the effective number of cases, calculated using the sampling weights. The effective number of cases is calculated as $y_{k}^{E}=n_{k}^{*}{\hat{p}}_{k}^{HT}$, with $n_{k}^{*}$ the effective sample size. This is modelled as
(10)$$\begin{array}{ccc} y_{k}^{E}|{\tilde{p}}_{k} & ∼ & \mathrm{Binomial}(n_{k}^{*},{\tilde{p}}_{k})\\ \mathrm{logit}({\tilde{p}}_{k}) & = & \beta_{0}+u_{k}+v_{k}. \end{array}$$

In this model, both the numerator and denominator are adjusted for the sampling design. This model is referred to as the effective sample size model (ES).

For all these hierarchical spatial smoothing method, both local and global information is borrowed amongst areas, via the use of the ICAR and independent random effects [16]. This ensures that predictions can be obtained for small areas in which no samples were taken. However, when the number of sampled spatial units is relatively low, the use of the ICAR model might become troublesome since the information that can be borrowed from neighbors is too sparse to gain reliable estimates [17]. In Section 4, we will investigate the behavior of these predictions.

### 3.3. Combining Survey Data with Auxiliary Summary Data

In previous models, predictions in non-sampled areas are based solely on the spatial random effect terms. This assumes that the only available information about the non-sampled areas is the spatial neighborhood structure of these areas. This is indeed correct if the survey at hand is the only source of information. In many cases, however, other sources of information are available as well. Often, summary data describing certain characteristics of areas are available. In this section, we propose a method to combine the survey data with auxiliary summary data.

In the survey, let $x_{ik}$ be a categorical variable available in the survey questionnaire which is related to the health outcome $y_{ik}$. The categorical variable has *G* possible values $x^{1},\ldots ,x^{G}$. In addition, assume that there is another data source which has summary information about this categorical variable in each area. Let $\pi_{gk}$ be the population fraction by which category *g* occurs in area *k*.

In the first step, we define the subgroup-specific outcomes, corresponding to the outcome variable in area *k* and group *g*. Depending on the model used, the outcome is defined in a different way. In the NB model $y_{gk}$ corresponds to the total number of cases in the sample in group *g* within area *k*, and $n_{gk}$ is the corresponding sample size. In the second step, we jointly model all the outcomes $y_{gk}$ (g=1,…,G and k=1,…,K) as
(11)$$\begin{array}{ccc} y_{gk}|{\tilde{p}}_{gk} & ∼ & \mathrm{Binomial}(n_{gk},{\tilde{p}}_{gk})\\ \mathrm{logit}({\tilde{p}}_{gk}) & = & \beta_{1}+\sum_{l=2}^{G}\beta_{l}I\left(x^{g}=l\right)+u_{k}+v_{k}. \end{array}$$

In this model, we allow the fact that the prevalence of the health outcome is different in the different subgroups, but we assume a common spatial smoothing amongst the different subgroups. In this way, information is shared amongst the different subgroups. In the third step, we use the auxiliary data as a post-stratification weight, in order to obtain a prediction of the overall prevalence at area-level *k*:(12)$${\hat{p}}_{k}=\sum_{g}{\tilde{p}}_{gk}\pi_{gk},$$

Note that the other models, discussed in previous section, can be adapted in similar way. If ${\hat{p}}_{gk}^{HT}$ corresponds to the HT estimator in group *g* and area *k*, then the empirical logistic transform and the arcsine square-root transform of ${\hat{p}}_{gk}^{HT}$, correspond to $y_{gk}$ in the LN and AS models. Similarly, $y_{gk}$ in the ES model is defined as the effective number of cases in group *g* within area *k*. Depending on the distribution of the subgroup-specific outcome, we assume that the outcome follows either a Binomial or a Gaussian distributions, as in (8)–(10). A joint model is formulated for the outcomes, assuming a shared spatial trend amongst the different subgroups. These subgroup-specific estimates are then combined, using the auxiliary data, to obtain an estimate of the overall prevalence.

## 4. Simulation Study

Two simulation studies are conducted. In the first simulation study, we investigate the impact of the number of missing areas in the context of a simple random two-stage sampling design. At the first level, districts are selected to be included in the survey. At the second level, individuals are sampled by simple random sampling. In the second simulation study, the impact of a complex sampling design in addition to missing districts is investigated.

### 4.1. Simulation 1: The Effect of Missing Areas

In this section we describe the setup of the simulation study to evaluate the performance of the spatial smoothing models when dealing with an increasing number of missing areas.

As building blocks for the simulation study we use the districts of Mozambique and their population sizes in 2008. We simulate a sample of 16,000 individuals according to a stratified random sampling design with proportional allocation. The samples are distributed across the 125 districts of Mozambique, proportionally to their respective populations sizes $N_{k}$. The corresponding design weight for each observation is therefore $w_{ik}=N_{k}/n_{k}$. Indeed, in this study, the survey does not have a complex survey design.

We simulate 100 times the outcomes for each of the individuals in the sample based on a Bernoulli model. The outcome for individual *i* in district *k* is simulated from
$$y_{ik}∼\mathrm{Bern}\left(p_{k}\right),$$
where
$$\mathrm{logit}\left(p_{k}\right)=\alpha +u_{k}+v_{k}.$$

The effects $v_{k}∼N\left(0,\sigma_{v}^{2}\right)$ are spatially unstructured effects and the effects $u_{k}$ are spatially structured effects that are sampled from a zero mean ICAR($\sigma_{u}^{2}$) model. These random effects (and consequently the prevalences $p_{k}$) are kept fixed across the 100 simulation runs. We simulate three different scenarios:(A1): we set $\sigma_{v}^{2}=0.15$ and $\sigma_{u}^{2}=0.03$;(A2): we set $\sigma_{v}^{2}=0.09$ and $\sigma_{u}^{2}=0.09$;(A3): we set $\sigma_{v}^{2}=0.03$ and $\sigma_{u}^{2}=0.15$.

The parameter α is set to 0. The assumed population proportions $P_{k}$ are presented, for each of the three scenarios, in Figure 2. In order to investigate the impact of the number of missing areas, for each simulation run, we randomly remove 5,10,15,…,70 areas from the sample. This corresponds to 4%,8%,12%,…,56% of the areas that are missing.

### 4.2. Simulation 2: The Effect of Auxiliary Data

In the second simulation, we evaluate the performance of the different small area estimators in the context of a complex sampling design. Data are simulated according to the sampling design of the PSIA survey. Models with and without the use of auxiliary data are compared. In this simulation, we use auxiliary information about the energy source that the households primarily use, which is a proxy for poverty of the individual.

For each individual sampled in the PSIA survey, we keep the design weight and its information with respect to use of energy as in the study. We simulate, for each individual in the survey, the outcome of interest:$$y_{ik}∼\mathrm{Bernoulli}\left(p_{ik}\right),$$
where we define $p_{ik}$ according to one of the following two assumptions
(*B*1) $\mathrm{logit}\left(p_{ik}\right)=\alpha +u_{k}+v_{k}$(*B*2) $\mathrm{logit}\left(p_{ik}\right)=\alpha +\sum_{l=1}^{7}\beta_{l}I\left({\mathrm{energy}}_{ik}=l\right)+u_{k}+v_{k}$,
reflecting a scenario in which the outcome depends only on the spatial structure (*B*1) and a scenario in which the outcome depends both on the spatial structure and the energy use of the individual (*B*2). Note that for scenario (B1)$p_{ik}\equiv p_{k}$, since no individual-specific information was used in the model. The spatial random effects $u_{k}$ and $v_{k}$ are given by either one of the three assumptions (A1), (A2), or (A3), reflecting a minor, mild, and strong spatial structure. The parameters α and β are chosen based on a preliminary analysis of the PSIA study (see Appendix A). For each of the scenarios, we simulated 100 data sets.

Each simulated data set is analyzed using four different model variations to construct the district-level estimates of the prevalences $p_{k}$. Each data set is analyzed using the different methods as described in Section 3. Table 2 summarized the different models. The first model (M0), is called the baseline model and corresponds to a model that does not take energy into account. All other models do use the auxiliary data on energy. Model (M1) assumes that the outcome probability depends on energy, but does not assume a spatial structure. Model (M2) assumes that the outcome only depends on the spatial structure, but not on energy. However a post-stratification is done in this model using the auxiliary variable. Note that this model is the same as model (M0) in the case of the naive binomial model, but not for the other model-based estimators (because of the use of the HT estimator as outcome in these models). Model (M3) does take into account both the spatial structure and energy. This is considered the most flexible model among models (M0) to (M3).

### 4.3. Summary Statistics

To evaluate the different estimators we compare the estimated squared bias and the estimated mean squared error (MSE). Denote by ${\hat{p}}_{k}^{\left(s\right)}$ the estimated proportion from the *s*th simulated data set, and $P_{k}$ the underlying true proportion. The statistics are then calculated as:$$\begin{array}{ccc} {\mathrm{Bias}}^{2} & = & \frac{1}{K}\sum_{k=1}^{K}{\left({\bar{p}}_{k}-P_{k}\right)}^{2},\;\mathrm{where}\;{\bar{p}}_{k}=\frac{1}{S}\sum_{s=1}^{S}{\hat{p}}_{k}^{\left(s\right)}\\ \mathrm{MSE} & = & \frac{1}{K}\sum_{k=1}^{K}\left(\frac{1}{S}\sum_{s=1}^{S}{\left({\hat{p}}_{k}^{\left(s\right)}-P_{k}\right)}^{2}\right). \end{array}$$

As the target is the estimand of the prevalence, the most obvious performance measures to consider is bias, precision and coverage of confidence intervals [18]. The bias measures whether the estimand is unbiased, which is desirable. An efficient estimator, i.e., an estimand with small variance, is also desirable, which is measured either by the precision of the estimand or the MSE (which is a combination of biasedness and efficiency).

We also calculated the coverage probabilities, whereby the nominal coverage is set at 95%:$$\mathrm{Coverage}=P\left({\hat{l}}_{s}\leq P_{k}\leq {\hat{u}}_{s}\right)=\frac{\sum_{k=1}^{K}\left(\sum_{s=1}^{S}I\left({\hat{l}}_{s}\leq P_{k}\leq {\hat{u}}_{s}\right)\right)}{S\cdot K},$$
where I(·) is an indicator function which return the value 1 if the true proportion $P_{k}$ lies between the lower bound ${\hat{l}}_{s}$ and upper bound ${\hat{u}}_{s}$ of the 95% confidence intervals for the design-based estimators or the 95% credible intervals for the model-based estimators, or returns the value 0 otherwise. Lastly, we calculated the average length of the 95% credible intervals of the prevalence estimates. Confidence intervals should have the property that at least 100(1-α)% of intervals contain the true value, which is measured by the coverage of the confidence intervals. We presented the results of all summary statistics for the in-sample and off-sample areas separately.

### 4.4. Simulation Results

#### 4.4.1. Simulation 1: Effect of Number of Missing Districts

Figure 3, Figure 4 and Figure 5 show the squared bias, MSE and coverage as function of an increasing number of missing areas. Figures on the left hand side correspond to the in-sample areas, i.e., areas for which samples are available; figures on the right hand side correspond to off-sample areas. Top panels correspond to scenario (A1) of a weak spatial structure, middle panels to scenario (A2) of a mild spatial structure and bottom panels to scenario (A3) of a strong spatial structure. Colors correspond to the different models (UNW, HT, NB, LN, AS and ES). Results are also given in Appendix A.

The squared bias for the design-based estimators (UNW and HT) in the in-sample areas are the smallest (Figure 3). This is expected, as these are considered unbiased estimators. Note that in this setting the UNW and HT estimators are the same, as the survey has a simple stratified random sampling design. The model-based estimators (NB, LN, AS, and ES) are all very similar, with slightly higher bias as compared to the design-based estimators. For all estimators (both design- and model-based), the bias in the in-sample areas is hardly affected by the number of missing areas, nor by the strength of the spatial structure. In the off-sample areas, however, a different result is observed. First of all, note that the design-based estimators are not available in the off-sample areas. Second, the bias increases with the number of missing areas. When the number of missing areas is small (less than 10 areas or 4% of the areas missing), the bias is small; but bias increases quickly before leveling-off when the number of missing areas is around 30 (24% of the areas missing). Third, the bias in the off-sample areas is smaller when the spatial trend is stronger. In this case, the information shared from neighboring regions is more informative for the prediction in the missing areas. Finally, all model-based estimators behave similarly in this setting.

In contrast to the bias, the MSE of the design-based estimators (UNW and HT) in the in-sample areas (Figure 4) is much higher. This is due to the instability (and high variability) of the design-based estimators when the sample sizes in the areas are small. This is why the design-based estimators are not recommended when the analysis is at a smaller geographical level. All model-based estimators behave similarly, and have small MSE in the in-sample areas, unaffected by the number of missing areas or the strength of the spatial structure. The MSE in the off-sample areas is higher compared to the MSE in the in-sample areas, but smaller than those of the design-based estimators of the in-sample areas. The MSE increases with the number of missing areas, but levels off when the number of areas is above 30. The MSE is smaller for stronger spatial structures, even if the number of missing areas is high (up to 56%).

The coverage for all methods (Figure 5) is excellent for the in-sample areas, and further from the nominal value (95%) for off-sample areas (however above 90% for all cases).

Based on these results, we conclude that the model-based estimators using spatial smoothing are very well capable of recovering the underlying spatial structure, even if the number of off-sample areas is large. The prediction in off-sample areas is better if the underlying spatial structure is stronger. Prediction in the in-sample areas is not adversely affected, despite the smaller overall sample size.

#### 4.4.2. Simulation 2: The Effect of Auxiliary Data

Figure 6, Figure 7 and Figure 8 show the squared bias, MSE and coverage of the different models ((M0)−(M3)), corresponding to the different simulation scenarios. Figures on the left hand side correspond to the in-sample areas, i.e., areas for which samples are available; figures on the right hand side correspond to off-sample areas. Top panels correspond to scenario (A1) of a weak spatial structure, middle panels to scenario (A2) of a mild spatial structure and bottom panels to scenario (A3) of a strong spatial structure. Within each plot, the left panel corresponds to scenario (B1) in which the true prevalence is spatially structured; the right panel corresponds to scenario (B2) in which the true prevalence depends on energy and the spatial location. Colors correspond to the different models (UNW, HT, NB, LN, AS, and ES). Results are also given in Appendix A.

We start with the results of the baseline model (M0), which does not take into account the auxiliary data and which was also used in the first simulation study. We observe again that the design-based estimators are slightly better than the model-based estimators with respect to bias in the in-sample areas. However, no estimation can be obtained using the design-based estimator in the off-sample areas. Also, the MSE of the design-based estimators is somewhat higher than the model-based estimators. The bias for the off-sample areas is higher, but slightly decreases as the spatial structure strengthens. Off-sample areas have a larger coverage than in-sample areas, due to the larger standard error by which the response is estimated. This is also confirmed by the average length of the CI, which is higher for all estimators in the off-sample areas. It can also be observed that using a strong spatial random effect in any simulation process for the response variable generally resulted in a better performance of the models for the off-sample areas. In the simulation scenario that depends on energy (B2), the bias slightly increases in the in-sample areas, though the impact is limited. The bias increases for the (M2) models, when comparing scenario (B2) to (B1). Since model (M2) cannot account for the effect of energy (simulated by scenario (B2)), this will result in an increase in bias for these model-based estimators. The difference is more pronounced for the off-sample areas, with higher biases and MSE when the underlying prevalence does depend on energy. We are interested in whether the use of auxiliary data can improve prediction, especially in the off-sample areas.

Model (M1) does take into account the auxiliary information, but neglects the spatial structure. This model is mis-specified and results in much worse estimates. The bias and MSE in the in-sample areas increases significantly based on the model-based methods, with major biases when there is a stronger spatial correlation. In the off-sample areas, the impact is less pronounced, except in the case of strong spatial trend (A3). In summary, the model-based estimators based on model (M1) under-perform for all summary statistics, for both the in-sample and off-sample areas. As such, one could conclude that the covariate energy alone is insufficient to produce reliable estimates in this context. The design-based estimators are constructed by averaging the area and energy-specific observed proportions. The bias and MSE of the design-based estimator increases, as a result of the sparseness in the data.

Model (M2) does only account for the auxiliary information via post-stratifying the estimated prevalences with respect to energy. It is the same as model (M0), but formulated at the energy-group level. Results are indeed similar to model (M0), but the model-based estimators outperform the design-based estimators in terms of bias and MSE in this case. Looking at the off-sample areas, there is a slight improvement across all summary statistics as compared to the baseline model (M0), when energy has a true impact on the prevalence. One could also observe that the results vastly improve when there is a stronger spatial effect in the data. This trend was reversed in model (M1) since the estimation model could not allow for the additional spatial variability in the data.

Model (M3) further reduces the bias in the off-sample areas, and in this case both energy and the spatial structure have an effect on the true prevalence. The results for the Bias^2^ and MSE perform better in model (M3) compared to the the baseline model in the off-sample areas. Bias, MSE and coverage are overall best for model the NB model, incorporating both energy and the spatial structure. Though the number of missing areas is large in this setting, the method works very well, and the use of auxiliary data improves estimation in the off-sample areas.

## 5. Data Analysis

In this section we analyzed the 2008 PSIA survey using the proposed methods. The outcome of interest was the proportion of school attendance in each of the 125 districts. However, since only 65 of these 125 districts were sampled, information from a secondary source was incorporated in the estimation process in order to obtain more reliable estimates. For the 2008 PSIA survey, the primary energy source of the household was considered as this would give an indication of wealth. Both a model without and with the auxiliary data were used.

### 5.1. Model without Auxiliary Data

Firstly, the PSIA data set was analyzed using the baseline model (M0). The results of the data analysis using the six estimators (UNW, HT, NB, LN, AS, and ES) are visualized in Figure 9. The unsampled areas are highlighted in white, with no estimation in these areas based on the design-based estimators. The model-based methods provide predictions for the proportion of school attendance in these areas, by incorporation of the spatial correlation between the areas in the estimation process. Overall, based on the model-based estimators, we conclude that the proportion of school attendance decreases gradually from south to north, in addition to some outlying areas in the center of Mozambique. Results of the different model-based estimators are very similar. The resulting trend is very smooth. Based on the simulation results, we know that we should look at these results with caution, as the trend is based only on the neighborhood structure. From the simulation results, it would be recommended to use the NB, LN, AS, or ES method.

### 5.2. Model with Auxiliary Data

In a second step, the PSIA survey was augmented with auxiliary information. The PSIA data was analyzed using a model which included the energy covariate and the two random effects in the linear predictor (M3). The final estimates for the six models are depicted in Figure 10. The design-based estimators seem to produce similar results as in Section 5.1, but with more area-to-area variation. As in the previous analysis, the NB, LN, AS, or ES method would be recommended for analysis of the data. These models again produce similar results overall. The information from the spatial correlation, augmented with auxiliary data from the energy variable produce a much more refined estimation of the school attendance proportions in Mozambique.

## 6. Conclusions

In this paper the impact of missing districts on small area estimation was investigated in the context of the 2008 PSIA survey. When implementing a survey for a country, it might occur that not all areas are included in the study. This will result in biased results when estimating parameters for the off-sample areas. Taking into account the spatial correlation between areas in the hierarchical models helps reduce this bias, but this is heavily reliant on the strength of this correlation. In order to gain reliable estimates for all areas, an extension to the classical hierarchical models was proposed: The inclusion of auxiliary data. Two simulation studies were performed to (1) quantify the effect of increasing numbers of missing spatial areas and (2) investigate the effect of auxiliary data on the spatial estimates, both for in-sample and off-sample areas.

When investigating the impact of the number of missing district in the first simulation setting, one could conclude that the estimates based on model-based method for the in-sample areas were unaffected, regardless of the number of missing districts. However, for the off-sample areas, the summary statistics for these estimates gradually declined until a plateau was reached at 30 missing districts (or 24% of the districts). On the other hand, a high spatial correlation between the districts did improve the performance of the off-sample areas, which was reflected in the summary statistics.

For the second simulation setting, we utilized the sampling scheme of the 2008 Mozambique PSIA survey. The variable energy was used as auxiliary information as it was considered to be an indication of wealth. Wealthier households are more apt in using modern sources of energy such as electricity and solar panels. Furthermore, we assumed that children from wealthier households had a higher probability of attending school. From the simulation, we concluded that results for the in-sample areas are largely not affected by the addition of auxiliary data. On the other hand, for the off-sample areas, the use of auxiliary data was a huge benefit if the model contained both covariate information and spatial random effects. One does have to be careful to ensure that the auxiliary variable(s) contains information which can complement the available data in the study. Design-based estimators might be very susceptible to auxiliary information, as it does not have access to the spatial correlation to smooth out this additional information.

In this paper, all hierarchical models were investigated at the level of the district, while the survey was designed to obtain reliable (design-based) estimates at a larger geographical scale only. If the number of unsampled districts is higher then the studied 56%, a spatial random effect at a higher geographical level could be needed [17]. This will model larger-scale spatial patterns and provide more reliable estimates if the amount of available data is sparse. This will be investigated in future research. A possible limitation of the proposed models is that only one source of auxiliary information was considered. As such, future research might include the investigation of multiple sources of auxiliary data.

## Figures and Tables

**Figure 1 ijerph-17-00786-f001:**
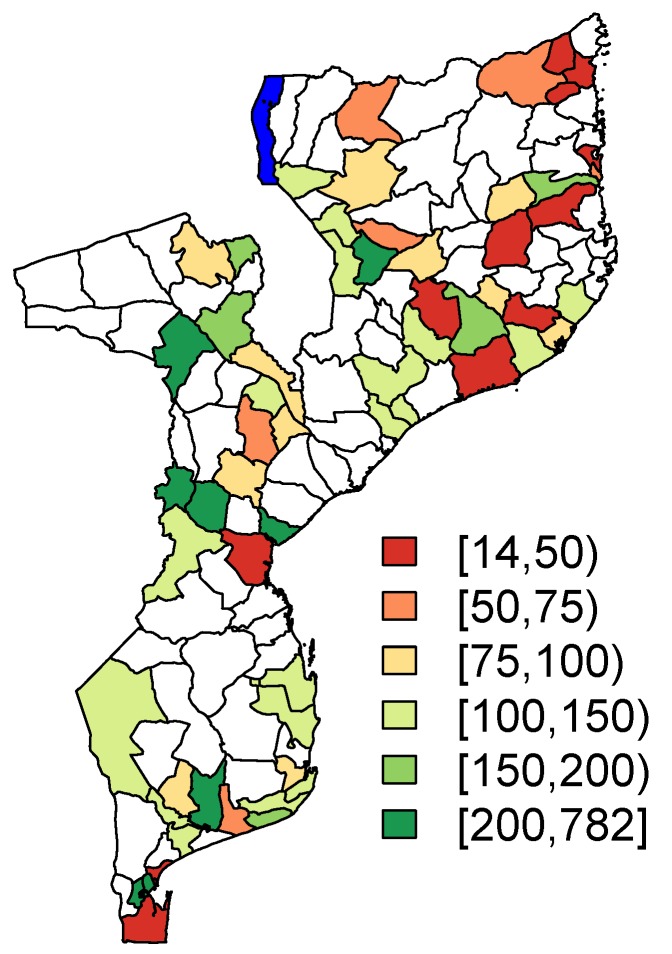
Sample size per district.

**Figure 2 ijerph-17-00786-f002:**
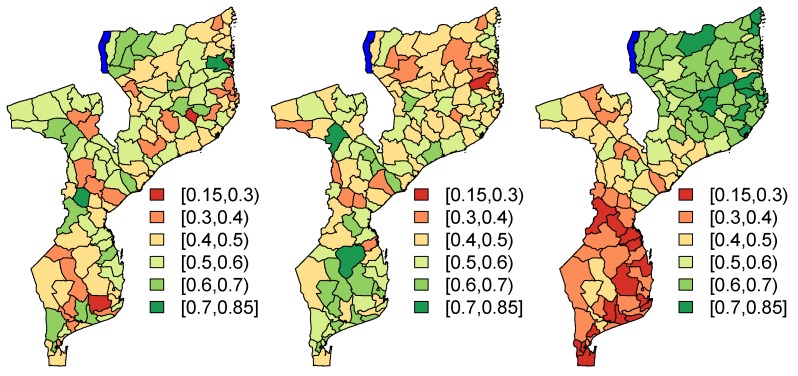
Plot of assumed population proportions $P_{k}$ for scenario (A1) (**left**), scenario (A2) (**middle**) and scenario (A3) (**right**).

**Figure 3 ijerph-17-00786-f003:**
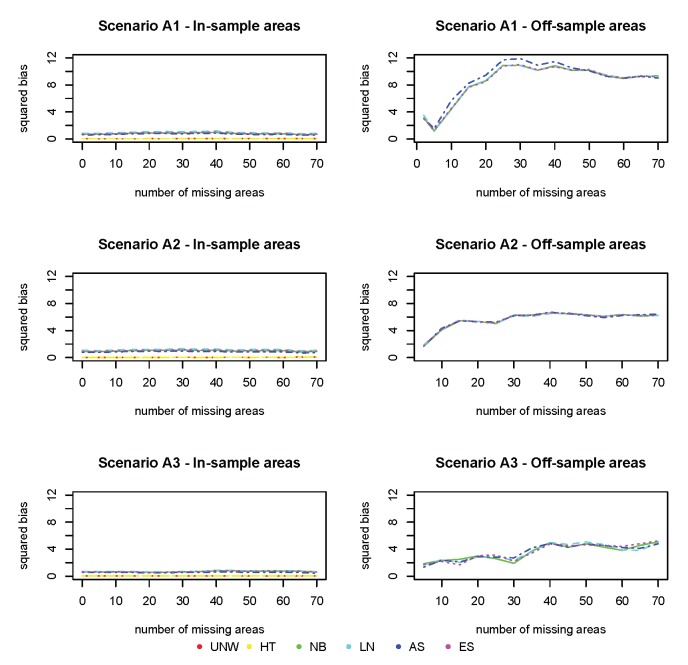
Simulation 1: Squared Bias ($\times 10^{3}$). UNW = unweighted estimator; HT = Horvitz–Thompson estimator; NB = naive binomial model; LN = logit-normal model; AS = arcsine model; ES = effective sample size model. (A1): $\sigma_{v}^{2}=0.15$ and $\sigma_{u}^{2}=0.03$; (A2): $\sigma_{v}^{2}=0.09$ and $\sigma_{u}^{2}=0.09$; (A3): $\sigma_{v}^{2}=0.03$ and $\sigma_{u}^{2}=0.15$.

**Figure 4 ijerph-17-00786-f004:**
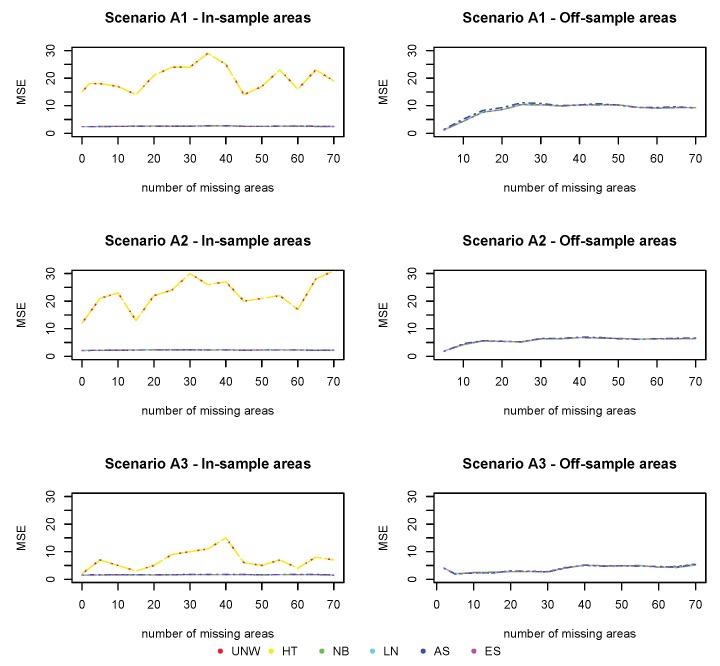
Simulation 1: Mean Squared Error ($\times 10^{3}$). UNW = unweighted estimator; HT = Horvitz–Thompson estimator; NB = naive binomial model; LN = logit-normal model; AS = arcsine model; ES = effective sample size model. (A1): $\sigma_{v}^{2}=0.15$ and $\sigma_{u}^{2}=0.03$; (A2): $\sigma_{v}^{2}=0.09$ and $\sigma_{u}^{2}=0.09$; (A3): $\sigma_{v}^{2}=0.03$ and $\sigma_{u}^{2}=0.15$.

**Figure 5 ijerph-17-00786-f005:**
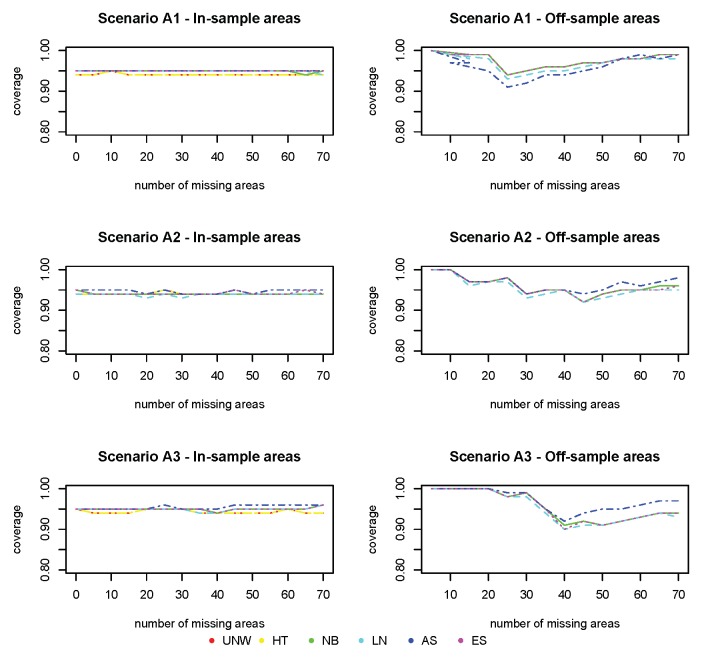
Simulation 1: Coverage probabilities. UNW = unweighted estimator; HT = Horvitz–Thompson estimator; NB = naive binomial model; LN = logit-normal model; AS = arcsine model; ES = effective sample size model. (A1): $\sigma_{v}^{2}=0.15$ and $\sigma_{u}^{2}=0.03$; (A2): $\sigma_{v}^{2}=0.09$ and $\sigma_{u}^{2}=0.09$; (A3): $\sigma_{v}^{2}=0.03$ and $\sigma_{u}^{2}=0.15$.

**Figure 6 ijerph-17-00786-f006:**
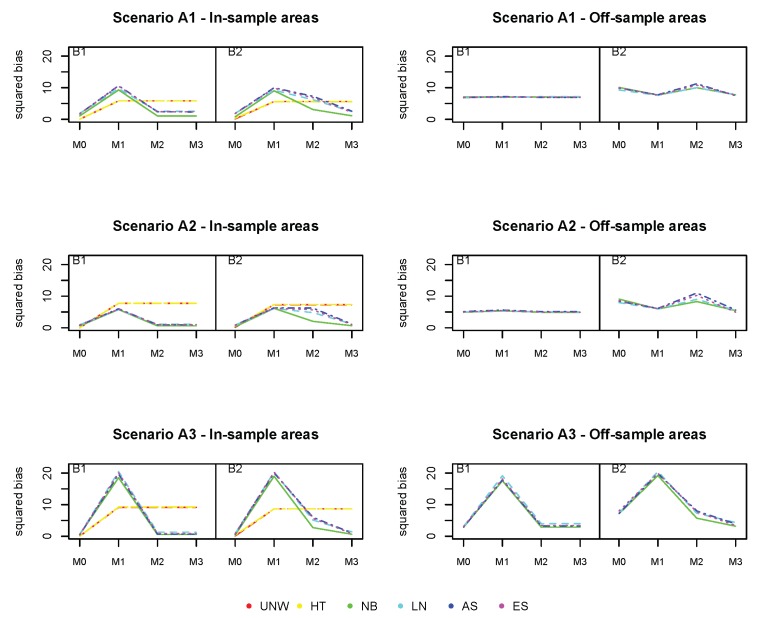
Simulation 2: Squared Bias ($\times 10^{3}$). UNW = unweighted estimator; HT = Horvitz–Thompson estimator; NB = naive binomial model; LN = logit-normal model; AS = arcsine model; ES = effective sample size model. (A1): $\sigma_{v}^{2}=0.15$ and $\sigma_{u}^{2}=0.03$; (A2): $\sigma_{v}^{2}=0.09$ and $\sigma_{u}^{2}=0.09$; (A3): $\sigma_{v}^{2}=0.03$ and $\sigma_{u}^{2}=0.15$
(B1): $\mathrm{logit}\left(p_{ik}\right)=\alpha +u_{k}+v_{k}$; (B2): $\mathrm{logit}\left(p_{ik}\right)=\alpha +\sum_{l=1}^{7}\beta_{l}I\left({\mathrm{energy}}_{ik}=l\right)+u_{k}+v_{k}$
(M0): $\mathrm{logit}(p_{k})$ = $\alpha +u_{k}+v_{k}$; (M1): $\mathrm{logit}(p_{gk})$ = $\alpha +\sum_{l=1}^{7}\beta_{l}I\left({\mathrm{energy}}_{gk}=l\right)$
(M2): $\mathrm{logit}(p_{gk})$ = $\alpha +u_{k}+v_{k}$; (M3): $\mathrm{logit}(p_{gk})$ = $\alpha +\sum_{l=1}^{7}\beta_{l}I\left({\mathrm{energy}}_{gk}=l\right)+u_{k}+v_{k}$.

**Figure 7 ijerph-17-00786-f007:**
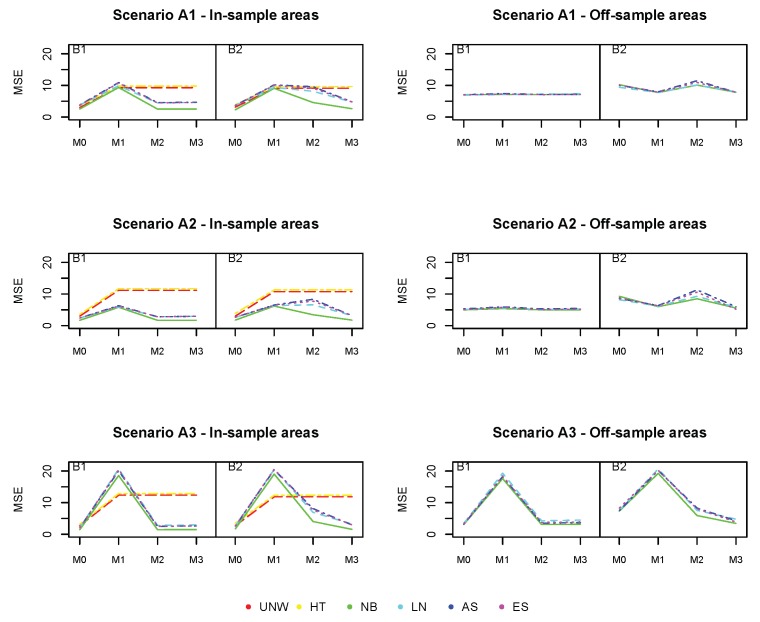
Simulation 2: MSE ($\times 10^{3}$). UNW = unweighted estimator; HT = Horvitz–Thompson estimator; NB = naive binomial model; LN = logit-normal model; AS = arcsine model; ES = effective sample size model. (A1): $\sigma_{v}^{2}=0.15$ and $\sigma_{u}^{2}=0.03$; (A2): $\sigma_{v}^{2}=0.09$ and $\sigma_{u}^{2}=0.09$; (A3): $\sigma_{v}^{2}=0.03$ and $\sigma_{u}^{2}=0.15$
(B1): $\mathrm{logit}\left(p_{ik}\right)=\alpha +u_{k}+v_{k}$; (B2): $\mathrm{logit}\left(p_{ik}\right)=\alpha +\sum_{l=1}^{7}\beta_{l}I\left({\mathrm{energy}}_{ik}=l\right)+u_{k}+v_{k}$
(M0): $\mathrm{logit}(p_{k})$ = $\alpha +u_{k}+v_{k}$; (M1): $\mathrm{logit}(p_{gk})$ = $\alpha +\sum_{l=1}^{7}\beta_{l}I\left({\mathrm{energy}}_{gk}=l\right)$
(M2): $\mathrm{logit}(p_{gk})$ = $\alpha +u_{k}+v_{k}$; (M3): $\mathrm{logit}(p_{gk})$ = $\alpha +\sum_{l=1}^{7}\beta_{l}I\left({\mathrm{energy}}_{gk}=l\right)+u_{k}+v_{k}$.

**Figure 8 ijerph-17-00786-f008:**
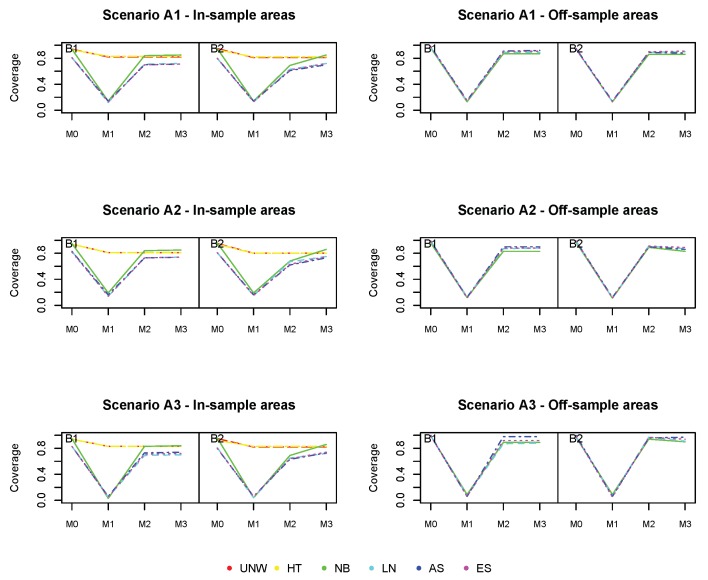
Simulation 2: Coverage probabilities. UNW = unweighted estimator; HT = Horvitz–Thompson estimator; NB = naive binomial model; LN = logit-normal model; AS = arcsine model; ES = effective sample size model. (A1): $\sigma_{v}^{2}=0.15$ and $\sigma_{u}^{2}=0.03$; (A2): $\sigma_{v}^{2}=0.09$ and $\sigma_{u}^{2}=0.09$; (A3): $\sigma_{v}^{2}=0.03$ and $\sigma_{u}^{2}=0.15$
(B1): $\mathrm{logit}\left(p_{ik}\right)=\alpha +u_{k}+v_{k}$; (B2): $\mathrm{logit}\left(p_{ik}\right)=\alpha +\sum_{l=1}^{7}\beta_{l}I\left({\mathrm{energy}}_{ik}=l\right)+u_{k}+v_{k}$
(M0): $\mathrm{logit}(p_{k})$ = $\alpha +u_{k}+v_{k}$; (M1): $\mathrm{logit}(p_{gk})$ = $\alpha +\sum_{l=1}^{7}\beta_{l}I\left({\mathrm{energy}}_{gk}=l\right)$
(M2): $\mathrm{logit}(p_{gk})$ = $\alpha +u_{k}+v_{k}$; (M3): $\mathrm{logit}(p_{gk})$ = $\alpha +\sum_{l=1}^{7}\beta_{l}I\left({\mathrm{energy}}_{gk}=l\right)+u_{k}+v_{k}$.

**Figure 9 ijerph-17-00786-f009:**
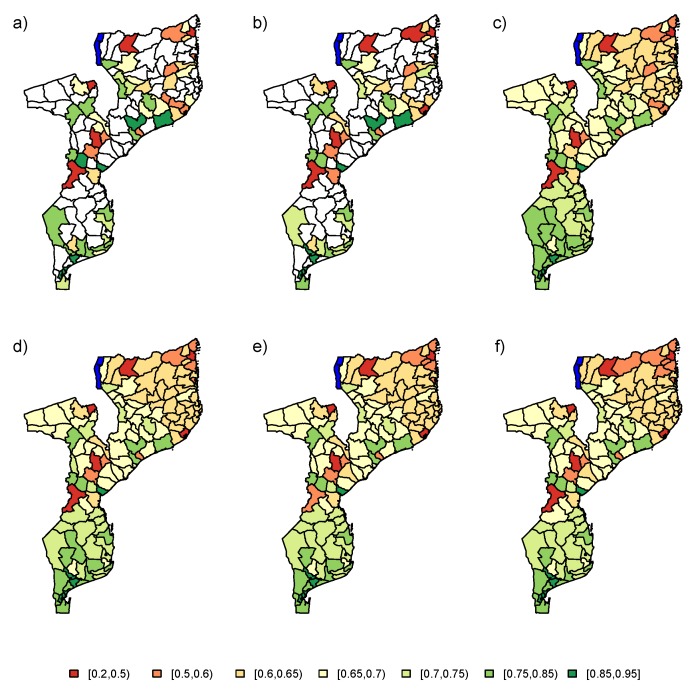
Maps for estimated school attendance proportions in 2008 PSIA study for (**a**) UNW, (**b**) HT, (**c**) NB, (**d**) LN, (**e**) AS, and (**f**) ES estimators. No auxiliary data was used in the modeling process. UNW = unweighted estimator; HT = Horvitz–Thompson estimator; NB = naive binomial model; LN = logit-normal model; AS = arcsine model; ES = effective sample size model.

**Figure 10 ijerph-17-00786-f010:**
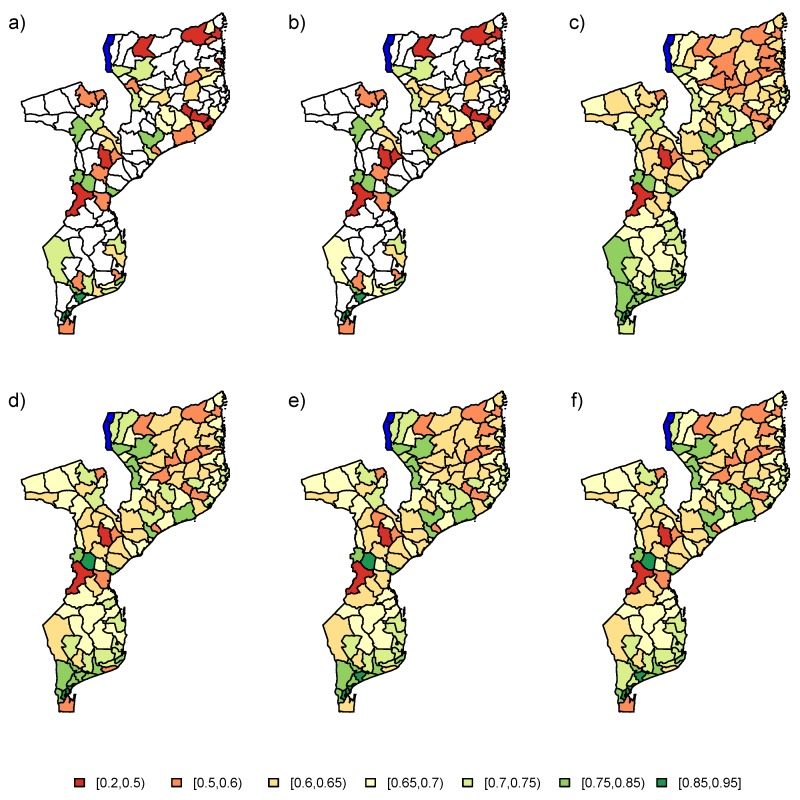
Maps for estimated school attendance proportions in 2008 PSIA study for (**a**) UNW, (**b**) HT, (**c**) NB, (**d**) LN, (**e**) AS, and (**f**) ES estimators. The information from variable “energy” was used as auxiliary data. UNW = unweighted estimator; HT = Horvitz–Thompson estimator; NB = naive binomial model; LN = logit-normal model; AS = arcsine model; ES = effective sample size model.

**Table 1 ijerph-17-00786-t001:** Population percentages for each energy source, averaged over the districts. The minimum and maximum population percentage is added between brackets.

Energy Source	Average Population Percentages (Range)
Battery	0.36% (0.01%–3.86%)
Candle	5.11% (0.28%–30.07%)
Electricity	4.68% (0.06%–62.84%)
Firewood	36.48% (0.06%–91.13%)
Gas	0.08% (0.00%–1.29%)
Generator/Solar power	0.38% (0.04%–1.83%)
Oil/Paraffin/Kerosene	52.19% (4.62%–89.76%)
Other	0.72% (0.05%–3.27%)

**Table 2 ijerph-17-00786-t002:** Overview simulation scenarios for Simulation 2. The first column is used to identify the matching simulation settings, specified in the second column.

*Model*	*Description*		
(M0)	$\mathrm{logit}(p_{k})$	=	$\alpha +u_{k}+v_{k}$
(M1)	$\mathrm{logit}(p_{gk})$	=	$\alpha +\sum_{l=1}^{7}\beta_{l}I\left({\mathrm{energy}}_{gk}=l\right)$
	${\hat{p}}_{k}$	=	$\sum_{g}{\tilde{p}}_{gk}\pi_{gk}$
(M2)	$\mathrm{logit}(p_{gk})$	=	$\alpha +u_{k}+v_{k}$
	${\hat{p}}_{k}$	=	$\sum_{g}{\tilde{p}}_{gk}\pi_{gk}$
(M3)	$\mathrm{logit}(p_{gk})$	=	$\alpha +\sum_{l=1}^{7}\beta_{l}I\left({\mathrm{energy}}_{gk}=l\right)+u_{k}+v_{k}$
	${\hat{p}}_{k}$	=	$\sum_{g}{\tilde{p}}_{gk}\pi_{gk}$

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
