# Peer review of "Spatial Modelling to Inform Public Health Based on Health Surveys: Impact of Unsampled Areas at Lower Geographical Scale"

_ijerph, 2020, doi:10.3390/ijerph17030786_

Round 1
Reviewer 1 Report
Referee report for the paper Spatial modelling to inform public health based on health surveys: impact of unsampled areas at lower geographical scale
Major comments:
Figures 3 to 10: Define briefly in a footnote what methods UNW, HT, NB, LN, AS and ES are. It is making it hard for the reader to look up those methods in Section 3.
Figures 9 and 10: Rather make use of sub-floats to distinguish between the methods, e.g. a) for UNW, b) for NB, etc.
Figures 3 to 8: In a footnote remind the reader in the figures themselves what scenarios ?1, ?2 and ?3 represent.
Figures 6 to 8: In a footnote remind the reader in the figures themselves what scenarios ?1 and ?2 represent.
Page 13, Line 266: In the text, explicitly call out those design-based estimators (which do badly). State which of the six methods they are.
Page 14, Line 272 … “but still smaller as compared to the design-based estimators”: The MSE of all six methods looks similar to me (Figure 4, “off-sample areas”). Would the authors care to explain?
Page 14, Line 272 … Add the word “missing” to “MSE increases with the number of missing areas, but levels off when the number of missing areas is above 30”.
Page 15, Lines 275 to 277. The findings are a bit over-explained. The confidence intervals are too conservative as well. I suggest you rather replace:
The coverage for all methods (Figure 5) is excellent, both for the in-sample as off-sample areas. The coverage in the off-sample areas is somewhat smaller if the number of missing areas gets larger, but the effect is limited, with coverage above 90% for all models.
with
The coverage for all methods (Figure 5) is excellent for the in-sample areas, and further from the nominal value (5%) for off-sample areas (however above 90% for all cases).
Page 16, Line 303: “impact is limited”. Can the authors justify this statement?
Minor comments:
Page 2, Line 32: “Most demographic and health surveys are designed …”
Page 2, Line 40: First use of abbreviation: Replace “SAE” with “small area estimation (SAE)”.
Page 2, Line 46: I suggest you replace “a lot of literature” with “many literature”.
Page 3, Line 52: I suggest you replace “an elaborate simulation study” with “an extensive simulation study”.
Section 2: What does the part in blue represent? It is described in Section 5. I think that text should rather be moved to Section 2.
Lines 201 to 203, Page 10: I think the index ? in ??? should be removed (i.e. ?? instead). Please check. I see three instances.
Page 10, Line 204: Replace “random effects are kept fix across” with “random effects are kept fixed across”.
Page 10, Line 204: What does ““random effects are kept fixed across” mean in the context of this simulation study?
Page 10, Lines 205 and 210: Replace “scenario's” with “scenarios”.
Page 11, Line 212: A comma needs to be added after “12%”.
Page 11, Lines 218 to 221: Again, ??? vs ??.
Page 12, Line 226. Rather say “see Table 7 …” Also, why isn’t this table presented first (Table 1)?
Page 12, Line 237: “This is the most flexible model” – Rather say: “This is considered the most flexible model among models ?0 to ?3”.
Page 13, Line 239: “To evaluate the different estimates we compare …” Estimators vs estimates? Maybe rather state “to compare models ?0 to ?3”.
Page 13, Line 254: This is expected, as these are considered unbiased estimators.
Page 13, Line 259: Typo. “results” vs “result”.
Page 14, Line 265 … “behave similarly in this setting”.
Page 14, Line 272: Change “but still smaller as compared to the design-based estimators” to “but smaller than those of the design-based estimators”.
Page 14, Line 273: Change – “Similar as with bias, MSE gets smaller when the spatial structure is stronger” to “the MSE is smaller for stronger spatial structures”.
Page 281, Page 15: Change “impacted” to “adversely affected”.
Line 284: Replace “scenario's” with “scenarios”.
Line 294: Replace “design-based estimators are slightly better as compared to the model-based estimators” with “design-based estimators are slightly better than the model-based estimators”.
Line 295: Replace “But” with “However”.
Line 296: Replace “And” with “Also”.
Line 296: Replace “higher as compared to the” with “higher than the”.
Line 297: Replace “The bias in the off-sample areas is higher” with “The bias for the off-sample areas is higher”.
Lines 297 and 298: Gets stronger? What about “strengthens” or “when there is a stronger …”?
Line 305: Change “We want to know whether the use of auxiliary data can improve prediction” to “We are interested in whether the use of auxiliary data can improve prediction”.
Line 309: Replace “increases a lot based on the” by “increases significantly based on the”.
Line 310: Gets stronger? See above comment.
Lines 326 and 327: Correction – “Bias, MSE and coverage are overall best for model”.
Line 336: Correction – “auxiliary data were used”.
Line 353: “previous paragraph”? Please be more specific.
Line 368: Replace “in simulation setting 1” with “in the first simulation setting”.
Reviewer 2 Report
In this manuscript hierarchical spatial smoothing models applied to demografic and health survey data on small areas are investigated.
It is necessary that the authors clearly highlight in the introductory section what is the purpose of their research and how they intend to make the comparisons between the investigated models in order to evaluate their performances.
In section 4.3 the authors must specify why Bias, MSE and coverage are a complete set of measurement of the accuracy and precision of the regression model. In addition, they must enter the description and formula of the Coverage probability and indicate the percentage of confidence.
English typos are present and must be corrected.
Round 2
Reviewer 2 Report
The authors have taken all my suggestions into account and have considerably improved the quality of their manuscript. I consider this paper publishable in the present form.